# A Novel Inspection Technique for Electronic Components Using Thermography (NITECT)

**DOI:** 10.3390/s20175013

**Published:** 2020-09-03

**Authors:** Haochen Liu, Lawrence Tinsley, Wayne Lam, Sri Addepalli, Xiaochen Liu, Andrew Starr, Yifan Zhao

**Affiliations:** 1School of Aerospace, Transport and Manufacturing, Cranfield University, Bedford MK43 0AL, UK; haochen.liu@cranfield.ac.uk (H.L.); l.tinsley@cranfield.ac.uk (L.T.); p.n.addepalli@cranfield.ac.uk (S.A.); xiaochen.liu@cranfield.ac.uk (X.L.); a.starr@cranfield.ac.uk (A.S.); 2ITA Labs, The International Tin Association, St Albans AL2 2DD, UK; wayne.lam@italabs.co.uk

**Keywords:** unverified electronic components, pulsed thermography, numerical simulation, feature selection, quantitative classification

## Abstract

Unverified or counterfeited electronic components pose a big threat globally because they could lead to malfunction of safety-critical systems and reduced reliability of high-hazard assets. The current inspection techniques are either expensive or slow, which becomes the bottleneck of large volume inspection. As a complement of the existing inspection capabilities, a pulsed thermography-based screening technique is proposed in this paper using a digital twin methodology. A FEM-based simulation unit is initially developed to simulate the internal structure of electronic components with deviations of multiple physical properties, informed by X-ray data, along with its thermal behaviour under exposure to instantaneous heat. A dedicated physical inspection unit is then integrated to verify the simulation unit and further improve the simulation by taking account of various uncertainties caused by equipment and samples. Principle component analysis is used for feature extraction, and then a set of machine learning-based classifiers are employed for quantitative classification. Evaluation results of 17 chips from different sources successfully demonstrate the effectiveness of the proposed technique.

## 1. Introduction 

Overviewing the global supply chain of electronic components (ECs), the severe threat of unverified or counterfeited products is omnipresently spreading and encroaching the benefit of legitimate trade. UnVerified Electronic Components (UVECs) and corresponding illegal trade rose from $461 billion to $509 billion between 2017 and 2019 [1,2], amounting to as much as 3.3% of the world trade of intellectual property. The UVECs can be defined as ECs with: (1) unauthorised copy; (2) inconformity to the original component manufacturer (OCM) design, models or standards; (3) unauthorised source; (4) broken, defective, or used OCM products; (5) fraudulent markings or documentation [3,4]. Without a doubt, those unverified components from untraceable sources are industrial unqualified and can lead to unstable and unreliable operational performance, causing unpredictable system malfunctioning and failure, operators’ injuries and catastrophic economic loss in manufacturing, transport, aviation, power generation, and cybersecurity every year [5]. Therefore, developing a reliable inspection technique is strongly demanded to equip and enhance the current capabilities of not just the asset owners but also the stakeholders in the entire value chain.

Figure 1 shows several unverified forms embedded in the supply chain and the corresponding defect causing them. Multiple defect types are related to the physical properties of ECs such as the material, internal structural anomaly (e.g., missing or broken die/wires, improper material, dimension and corroded or oxidised mould packaging, etc.) and electronic malfunctions (e.g., current leak, transistor variation, improper electrical heat emission etc.), while others are related to external markings and trade paperwork (e.g., sanding marking, forged shipping label or paperwork, etc.). In response, a few inspection techniques aimed at testing the interior and exterior integrity, and electronic functionality, have emerged to screen electronic components in the supply chain, which can be broadly categorised as either physical or electrical tests [6,7,8]. However, subjecting to the limitations in inspection speed, cost and capability, stakeholders normally have to deploy multiple inspection methods to perfect exhaustive screening for different components and anomalies inside ECs (internal structure of typical dual in-line package EC shown in Figure 2) [9,10].

For instance, as an exterior physical inspection, random spot-checking results in a high uncertainty regarding the pass/failure product ratio due to the inappropriate selection in a large number of batch testing [11]. X-ray imaging and THz testing are capable to analyse the accurate internal physical properties such as wires, circuit joint and package material, but cannot be widely used due to them either being destructive with low throughput and a high capital cost per inspection unit, or overly sophisticated methodology for unskilled operators [12,13,14]. Besides, electrical testing like burn-in testing, current leakage testing and function verification are incapable of detecting structural defects and aged components [15]. Therefore, an intuitive, non-contact, high-efficient and low-cost non-destructive inspection technique is highly demanded to deploy the first stage large scale inspection efficiently for reducing the proliferation of UVECs.

Among the diverse non-destructive testing (NDT) inspection methods, pulsed thermography (PT) is especially attractive for the inspection of UVECs because it allows the inspection of a significant number of components simultaneously, facilitating a rapid and relatively low-cost inspection per part due to its rapid, non-contact inspection manner and its ability to produce intuitive inspection images [16,17]. As it is a thermal-based penetration inspection technique, the inner material and structural defects in ECs can be interrogated by reconstructing the thermal features of captured thermal decay profile and image. Although quantitative characterisation of damage/defect in uncomplicated structure has been well studied and proven to be effective [18,19,20,21], as a new inspection technique against UVECs, the thermal response and corresponding detectability using PT for UVECs are still unknown.

Furthermore, a systematic investigation combined together with a simulation-based approach, can conveniently provide predictive insights of the PT detectability for the specific defect types in UVECs, thus remaining unascertained and scientifically worthwhile. This research addresses the challenge to identify UVECs with physical defects including the deviation of internal structures and material properties in the die, lead frame and the mould packaging which are typically found in the supply chain of ECs. For instance, components with undersized die and over-simplified lead frame from disqualified manufacturing are reported to be short-lived and incapable in electronic option [22]. Besides, unsourced, low-cost and unqualified mould materials have been found in chips and have failed in long-term thermal fatigue cycles and ageing tests [23]. It is therefore essential to clarify not only the inspection feasibility, but also which variability can be ascertained in PT thermography.

In the thermography simulation investigation for ECs, several studies have investigated and predicted the thermal response of the die part for electronic performance evaluation [24,25]. However, the model only focuses on internal electrical heating on the die, where steady-state thermal signals measured on dies are captured. The thermal dynamics of mould packaging, lead frame heated by pulses have not been studied, therefore modelling of the real ECs whole structures and temperature transfer through chips is essential. As a new technique applied to ECs, the inspection capability for specific internal deviation such as die and lead frame is still unknown. The inspection feasibility of batch-chip inspection deployment and influence of ECs surface condition remain unclear [26]. In addition, for the large quantity inspection scenario, variation and uncertainty of heating uniformity, external pins and surface marking between chips have a strong impact on the test accuracy which should be taken into account. Compared to the deterministic decision-making, AI-based decision-making strategy can provide a rapid and robust detection against uncertainty and variation between ECs [27,28,29]. The corresponding strategy can seamlessly set a powerful foundation for detection decision making for this research.

## 2. Methodology

In this study, we propose a novel inspection technique consisting of numerical modelling of PT, experimental verification to answer the aforementioned challenges and thus develop a unique AI (machine learning) powered digital twin system to identify the UVECs with internal structure deviation from multiple manufacturers. Figure 3a,b presents the principal of PT inspection for EC and an overview of the methodology of the proposed inspection technique for UVECs, respectively.

For the first unit, a model-based finite element model (FEM), based on the internal structure of verified components informed by the X-ray inspection, is established to predict the thermal behaviour of ECs in PT. With such a simulation tool, we can obtain not only the thermal signals of different types of variation of physical properties of an EC but also a pre-inspection foundation for effective feature selection in the scope of thermal image processing. For the second unit, a PT inspection system is established to mimic the simulation unit by identifying the component variability in both single-chip inspection mode and batch-chip inspection mode.

The inspection results can validate not only the performance of the simulation, but also validate the simulation modelling (e.g., material thermal attributes refinement, interface adjustment between layers and component surface condition approximation), together with quantifying the noise level in inspections. With the support of feature extraction and selection, the optimal features will be selected and evaluated to distinguish UVECs from the verified components using machine learning approaches. Both the single-chip and batch-chip testing modes aim to demonstrate the feasibility and flexibility of PT for UVECs and the enhanced accessibility to the supply chain. Benefitting from the flexibility of simulation and applicability of experiments, this bilateral Intelligent inspection system can provide inspectors not only a powerful tool to detect UVEC with different defect types but also a confident quantification of its detectability.

### 2.1. Electronic Components

To achieve the proposed technique, this research starts with a common type of electronic chip: operational amplifiers (OpAmps). A total of 17 OpAmps were collected from various trade markets to investigate their thermal responses in PT. All samples are common dual in-line package chips and are designed for the same electronic purpose (e.g., UA741CN). They have almost identical external appearance except for surface markings. It should be noted that this study has no interest in investigating the surface labelling. To reveal the internal structure of these chips, X-ray inspection was adopted to obtain their lead frame and die details. Figure 4 presents the exterior dimensions, surface markings, and X-Ray images of 17 samples, which can be categorised into three groups according to the surface markings.

Firstly, as shown by surface marking, chips of group A (from ‘001’ to ‘004’) and B (from ‘005’ to ‘010’) are from the same origin marked by “CHN” but those of Group C (from ‘011’ to ‘017’) are original from “MAL”. The markings of “UA741CN” suggest same electronic function. The ECs of Group B and C were tested by the electronic function testing and proven to be malfunctional or broken, while others in Group A were verified as sound ones. Secondly, all chips were tested using the same X-ray inspection and the result of Group A is considered as the ground truth of the internal structure. The internal structure of the lead frame and die in Group A and B are consistent, while those in Group C exhibit variations in the lead frame layout, die position and size. The die of ‘011’, ‘013’, ‘015’ and ‘017’ appears to the right half of the chip while the die of ‘012’, ‘014’ and ‘016’ are positioned towards the left. Additionally, based on the intensity in X-Ray images, chips in Group C have different mould materials, which is a common variability between manufacturers [30]. Based on these preliminary information set, the strategies from both simulation modelling and experimental inspection are discussed below.

### 2.2. FEM Numerical Modelling

In this research, the electronic components are in a typical dual in-line packaged type which is mainly composed of multiple layers composed of three primary materials: mould, lead frame and die. To simulate the real structure of the EC, the X-ray images of verified chips in Group A (see in Figure 4) are referenced to provide an accurate layout of the internal lead frame and die. According to the X-Ray images, the layout and dimension of the lead frame and die dimension are initially transferred to a binary mask. Then by mapping the mask onto the specific layers of the model, the real internal structure is simulated in the virtual environment. As a result, a 3D model is established with three layers with different materials, shown in Figure 5, including the mould compound layers, the lead frame layer and the die layer. Homogenous materials were used for the lead frame and die layers, and the thermal property of each layer is approximated by referring [21], the details of which are shown in Table 1. The thicknesses of the upper, lower mould and lead layer are 1.3, 1.5 and 0.2 mm, respectively. The 0.1 mm die is sitting on the plate area of the lead frame. The model is surrounded by air material element, and the pins connected to the lead frame is ignored for simplification. The whole model is discretised into 160 × 128 × 50 units using Hexahedral iso-parametric element.

The transient thermal response of PT inspection can be simulated by the heat conduction equation shown in Equation (1):(1)$$\kappa {\left\{\nabla \right\}}^{T}\left\{\nabla \right\}T-\rho c\dot{T}=-{Q|}_{\Gamma }$$
(2)$$\left[K\right]\left\{T\right\}+\left[C\right]\left\{\dot{T}\right\}=\left\{Q\right\}$$
where *T* and {∇} denote the temperature and gradient vector and *–Q|_Γ_* is the excitation heat flux applied on the surface. *κ*, *ρ* and *c* denote the heat conductivity, density and specific heat of material respectively. Assuming no heat loss, the temperature of the tested sample can be conducted by the FEM governing Equation (2), where [*K*], [*C*] and {*Q*} stand for coefficient stiffness matrix of conductivity, volumetric specific heat and excitation heat source respectively. To solve Eq. (2), a time-domain integration solver was employed in Eq. (3), where *ζ* = 0.3 is a convergence parameter that controls the calculation stability and accuracy. Additionally, according to the thermal condition of PT inspection, constant thermal properties are used, and the thermal expansion effect is neglected. Besides, the non-adiabatic surrounding condition is adopted with the EC model being assumed to be surrounded by air elements [31,32,33]:(3)$$\left[\left[K\right](1-\zeta )+\frac{\left[C\right]}{\Delta t}\right]{\left\{\dot{T}\right\}}_{t+\Delta t}={\left\{Q\right\}}_{t+\Delta t}+\left[\frac{\left[C\right]}{\Delta t}-\zeta \left[K\right]\right]{\left\{T\right\}}_{t}$$

In PT, a homogenous flash is used to heat the top surface of the sample. Considering the small size of ECs, the heat distribution in a single EC can be assumed to be uniform. The temporal pulse wave of heating is shown in Figure 5. The pulse peak of heat flux density applied on the surface elements is 5.0 × 10^4^ W/m^2^. The pulse duration lasts for 0.02 s when the flash intensity decays from 0.002 s. The profile of pulse duration is simulated according to the measured data instruction of the commercial pulse lamps used in the experiment.

### 2.3. Feature Extraction of Thermal Signals

To reveal the important features from the thermal response that could be caused by physical deviations of components, Principal Component Analysis (PCA) is used for the extraction of the spatial and temporal information from a thermographic matrix formed by an image sequence. PCA has been shown to yield high levels of thermal contrast for underlying structural damage resulting in satisfactory detectability compared to conventional thermographic procedures [34].

Singular value decomposition (SVD) is a powerful method to compute the principal components (PCs). In general, the SVD of an *M* × *N* matrix *A (M > N)* is a factorisation of the following equation:(4)$$A=URV^{T}$$
where *U* is an *M × N* orthogonal matrix, *R* is a diagonal matrix (with the non-negative singular values of *A* on the diagonal) and *V^T^* is the transpose of an *N × N* orthogonal matrix. The thermal image sequence is arranged in the form of a timestamp as the column-wise and the pixels as the row-wise of *A*. And the columns of decomposed *U* comprise the orthogonal statistical modes known as empirical orthogonal functions (EOFs) which describe the spatial variation in the thermograms. Likewise, PCs, which describe the temporal variation, comprise the rows in matrix *V^T^*. Commonly, the first three orders of EOFs and PCs provide sufficient description of data variability in the space and time domain.

For the PT inspection of multiple ECs or batch-chip mode, the uncertainty of heat uniformity, homogeneity, energy absorption and surface condition would lead to inconsistency in temperature profile such as peak value, peak time and steady-state temperature. Detection analysis for variability only in the temperature field will still be insensitive to the target variation even with accurate data calibration. However, PCs have the potential to mitigate those uncertainties and focus on the principal transient variation, which can effectively identify the variation in material properties, die size and lead frame [26]. In this research, the absolute values of the 2nd and 3rd PCs are proposed to characterise the thermal behaviour of each chip.

### 2.4. Experimental Testing Unit

In this study, a PT-based prototype (illustrated in Figure 6) was established, where a short and high-energy light pulse was projected onto the sample surface through two commercial flash lamps. The temperature measuring device was an FLIR A615 series IR camera with a long-wave (7–14 µm) uncooled focal plana array. The top surface temperature of ECs was captured with a maximum 640 × 512 pixels resolution and a framerate of 25 Hz. Both the camera and the lamps were synchronised and controlled by a PC to perform an automated inspection.

A close-up IR lens with a magnification factor of x2.9 was adopted to achieve a 50 μm spatial resolution and a field of view of 32 × 24 mm. The maximum energy applied in the inspection was approximately 1.4 kJ over an area of 500 mm diameter. The samples were placed with their surfaces perpendicular to the IR camera’s line of sight at 85 mm from the lens. To reduce the uncertainty of heat uniformity, the samples were placed at the centre of the camera view. Both lamps heated the samples with a 0.01 s pulse duration and maximal energy.

Two experiments were conducted to evaluate its performance of the single-chip inspection mode and batch-chip inspection mode, respectively. In the first experiment, the selected 17 chips (shown in Figure 4) were inspected one by one. Each thermal image sequence was captured at 25 Hz and 8 s duration, where 200 frames were captured for analysis. To exclude the influence of external pins and boundary, a region of interest (ROI) for each sample was selected covering 90% area around the centre of the chip. Between each inspection, a five-minute interval was implemented to ensure the lamps and camera have reached equilibrium temperature. In the second experiment, an inspection of four chips from Group A and C simultaneously was conducted to verify its deployment feasibility in the batch-chip testing. The experimental parameters and the selection of ROI are kept the same as the first experiment.

## 3. Results

### 3.1. Simulation Results

To investigate the transient temperature transfer, the verified component model and variation models were established and simulated. Figure 7 presents the temperature distributions of the verified component model (shown in Figure 5) at 0.04, 0.08, 0.12 and 0.16 s after the flash.

It also shows the results of a vertical and horizontal slice through the chip and along the lead layer, which presents a clear temperature distribution and penetration through the mould to the die layer. With the time elapsing, the heat quickly passes through and results in an obvious temperature pattern in accordance with the lead frame and die layout. The top-surface temperature signals against time were then extracted, and random white noises were pixel-wisely added with a signal-to-noise ratio (SNR) of 60 dB. The added noise level was selected based on the developed experimental testing unit, which was tested and pre-analysed by the thermographic signal reconstruction (TSR) algorithm for ECs [29]. After obtaining the simulated results, the ROI containing 90% of the surface area around the centre is selected for feature extraction based on PCA.

In order to observe the thermal features of anomaly deviation in UVEC, the single variable deviation of simulation model provides the most efficient tool to figure out the relation between them. Three types of parameter deviation were simulated (Table 2) and corresponding the 2nd and 3rd PCs were extracted. They were consistent with the difference observed in three groups of samples in Figure 4. As shown in Table 2, for the variability type of die size and material conductivity, four models marked with the deviation percentage of the parameter were simulated, where the reference model was also added for comparison. The variability of die size varies from 30% of the reference to 250%, which produces about 1mm^2^ size variation for two successive models. The mould material heat conductivity varies from 110% of the reference to 70%. In addition, two layouts of the lead frame including the reference in Group A and ‘011’ in Group C were simulated.

Initially, Figure 8a,b show the comparison of the 2nd and 3rd PCs with different values of die size. It can be observed that the peak of the 2nd PC curves appears at around 0.5 s after the flash with a lower amplitude due to the increase of die size. And the peak value and peak time of the 3rd PC profiles (around 2–3 s after the flash) appears lower and later monotonically. Furthermore, the 2nd PC peak seems to be more sensitive than the 3rd PC to the change of die size. Secondly, Figure 9a,b show the comparison of the 2nd and 3rd PCs with different values of material conductivity. The peak amplitude of the 2nd PC decreases significantly in a monotonical trend following the decrease of conductivity. And the peak amplitude and peak time of the 3rd PC become lower and later in a more dramatic manner, which shows opposite monotonicity to the trends of the die size variation. Viewing the feature trends of die size, it represents that the bigger die has the smoothing effect of the 2nd PC peak and delay the 3rd PC peak. For the mould material deviation, the feature trend indicates the lower conductivity mould will maintain the heat flow longer resulting in similar smoothing and delay effect. Additionally, it is observed that these two features are more sensitive to the change of mould material than that of the die size. This is probably because the mould material occupies most of the EC, and the heat diffusivity of mould is much lower compared to the small die. Thirdly, from the simulation results of lead frame deviation shown in Figure 10, both the 2nd and 3rd PCs can identify the difference in the layout. Although there is only lead frame changing in this comparison, the thermal behaviour variation still is observed by these two features in terms of peak value and peak time. Based on the above observations, the peaks of both 2nd PC and 3rd PC, therefore, can be effective features to identify these three variations in ECs.

### 3.2. Experimental Results

This section presents the experimental results using the developed testing prototype to verify the selected features mentioned above under the single-chip inspection mode. Figure 11 shows the first temperature frame (0.04 s after the flash) of all 17 chips. It should be noted that they were captured one by one, and the location in the scene and other experimental parameters are identical. From this figure, it is suggested that there is no obvious pattern in the raw IR images that can reliably distinguish these three groups.

Even for the same group, such as Group A, the IR images suffer the uncertainty of heating uniformity and emissivity. The reason is mainly caused by the uncertainty of the surface emissivity of samples which results in different temperature rise after the pulse. Besides, the uncertainty of surface homogeneity leads to a diversity of thermal spatial distribution in the first frame. According to the ground truth from the X-ray images, two comparisons were implemented to verify the proposed technique. The first one is between Group A and ‘012’, ‘014’ and ‘016’ of Group C. They have the same lead layout but are from different manufacturers. This comparison focuses on the deviation of mould material and the die size. The second one compares the chips in Group C with two different lead frame layouts.

From the PCs profile shown in Figure 12a,b, where solid curves and dotted curves indicate the two groups, it is suggested that the deviation of mould material and die size can be effectively reflected by the peak of 2nd and 3rd PCs. The behaviour of features is consistent with the simulation results and in return proves the effectiveness of the simulation unit. For the 2nd PC, the chips in Group A (solid curves) have significantly higher peak values around 0.5 s after the flash than those in Group C (dotted curves), meanwhile, the peak times consistently appear earlier. For the 3rd PC, the chips in Group A have slightly higher peak values around 1–2 s than those in Group C, meanwhile, the peak times appear significantly earlier. These observations suggest that Group C has a larger die (twice bigger than Group A) and mould material with a lower conductivity than Group A, which can be confirmed by the X-ray images in Figure 4. Figure 13a,b show the features of the second comparison, where the chips with the right lead layout (solid curves, indicated as Group C-2) are distinguishable with the ones with the reference lead layout (dotted curves, indicated as Group C-1). The chips in Group C-2 have significantly higher peak values and slightly earlier peak time in the 2nd PC than the chips in Group C-1. Meanwhile, for the 3rd PC, they have similar peak values but Group C-2 peaks significantly earlier than Group C-1(about 0.5 s earlier). A quantitative comparison of the peak features among the three groups for these two comparisons are shown in Figure 14. It seems that the best feature to classify these groups are the 2nd PC peak value, then 3rd peak time, 2nd PC peak time and 3rd PC peak value.

Based on the selected four features shown in Figure 14, different feature combinations in the scatter plot are presented in Figure 15, where the chips in Group A, B, C-1 and C-2 are plotted in red, blue, black and green colour, respectively. The scatter distribution of the 2nd PC peak value and 3rd PC peak time (the top right graph) presents the most promising trend for the classification of these four groups. The second-best selection is the feature combination of the 2nd PC peak value and 3rd PCs peak value. Based on the best feature selection, it is suggested that the PT inspection can effectively distinguish the verified chips from Group A (red) and unverified ones from Group C (green and black). Moreover, it is also capable to distinguish the two groups with different die sizes and lead frame layouts, denoted by black and green dots. However, it is relatively difficult to distinguish the malfunctional chips in Group B (blue dots) from the verified chips in Group A (red dots). This is because those in Group B are materially and structurally consistent with ones in Group A. The electronic malfunction might be led by a short circuit in the die or poor contact in wires.

To quantitatively evaluate the classification performance, machine learning approaches are then applied to the best feature combinations, where the 5-fold cross-validation was used. Table 3 shows the classification accuracy between Group A, B and C based on the 2nd PC peak value and 3rd PC peak time using three classical clustering algorithms (K-nearest neighbours (KNN), linear discriminant (LD), support vector machine (SVM)). The results demonstrate that it has a 100% accuracy to distinguish UVECs in Group C with ones in Group A, while the accuracy is reduced to 70% for Group A and B. Even for the three and four classes, the classification accuracy is still relatively high (>88%) where different types of deviation are presented.

## 4. Discussion

This paper proposed a feasibility study to develop an effective tool for identifying UVECs with physical defects, such as internal structural and material variabilities. It is expected that the PT inspection cannot offer the similar high-resolution of internal structure as the X-ray inspection due to its physical limitations. However, this technique aims to act as a first-stage screening tool in a non-destructive, time effective and automatic manner. Further functional testing can be followed on the suspected ones to balance the time and confidence level of the inspection. Depending on the cost of EC going into the production line, some industries might look to have some sort of inspection capability in-house to screen components before they go to assembly. However, this is usually in the form of random sample checks on single components selected from batches of components and it is almost impossible at this stage to check all components cost-effectively. Another emerging problem is that the cost to inspect a component can be much higher than the value of the component itself especially with the technological advancements in manufacturing. Therefore, this technique is particularly important for low-value high critical components that directly relate to the reliability of the system performance.

This research has evaluated the performance under the single-chip inspection mode, where only one chip is inspected at one time. Although such an approach mitigates the influence of heat distribution at different locations, it is not time effective. Inspection under the batch-chip mode will be attractive for the supply chain due to its robustness and rapid turnaround times. Figure 16a presents the first temperature frame of four selected chips (3 from Group A and 1 from Group C) and Figure 16b,c plot the corresponding PCs feature profiles.

The chip of ‘013’ from Group C exhibits a significant difference with the ones from Group A in terms of the peak features of both PCs. This observation demonstrates the applicability of the proposed technique for batch inspection. Furthermore, it should be noted the PCs are orientation-free as the space information has been removed. Such a characteristic is helpful when a large number of chips are required to be inspected at the same time.

## 5. Conclusions

This paper reports a new UVEC inspection technique that includes the FEM simulation and experimental prototype, following the digital twin methodology. The simulation is able to model the complex internal structure of ECs, informed by X-ray images, and effectively predict the surface thermal behaviour after application of instantaneous heat. The results suggest that the 2nd and 3rd peak of principal components of temperature delay profile can provide a sensitive and robust indication for the deviation of die size, lead frame layout and mould material properties. This research finding has been confirmed through using the same data analysis method for the experimental data, which were collected by a new established dedicated pulsed thermography prototype. With the support of machine learning-based classifier, quantitative results suggest that the proposed technique can effectively identify the unverified components with certain robustness considering the variation of verified components. We also demonstrate that this technique can work under both single-inspection and batch-inspection modes, which offers the deployment flexibility to the supply chain. The proposed technique can act as a powerful screening tool, after which other NDT techniques can be conducted on suspicious chips to improve the efficiency of UVEC inspection.

Future work in this area will extend the inspection scale of UVECs to establish a feature signal database of different types of UVECs. It will help both improving the simulation modelling of details in EC, upgrades to the versatility of the inspection system and significantly enhance the AI decision making strategy. In the feature selection approach, the EOFs in PCA will contain more spatial information of the ECs internal structure like lead and die layout which could enhance the inspection visualisation. It will be investigated carefully in future.

## Figures and Tables

**Figure 1 sensors-20-05013-f001:**
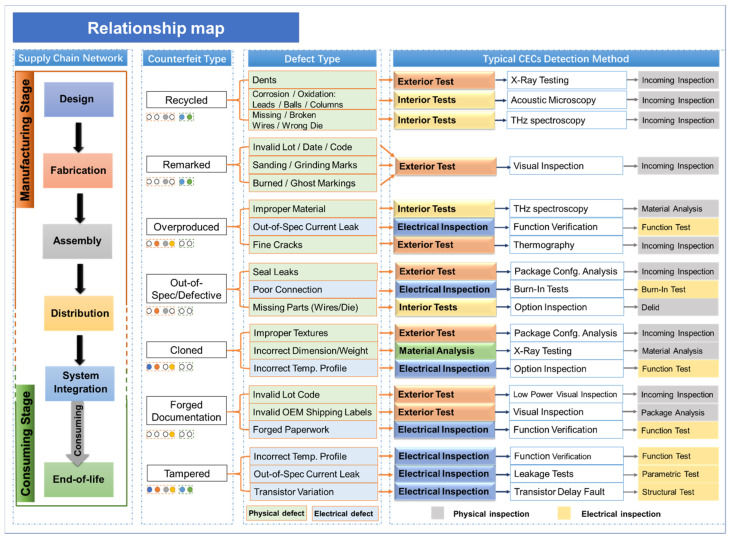
UVEC types and detection methods in the supply chain.

**Figure 2 sensors-20-05013-f002:**
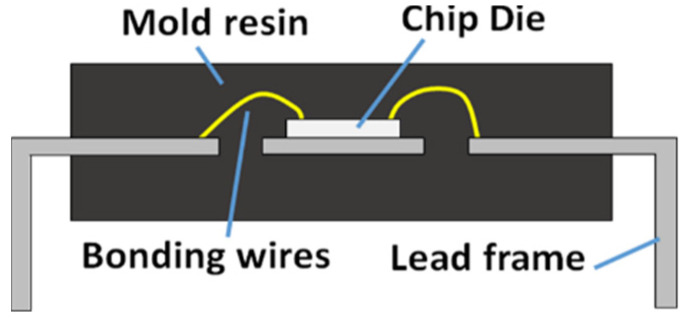
The simplified structure of a typical dual in-line package EC.

**Figure 3 sensors-20-05013-f003:**
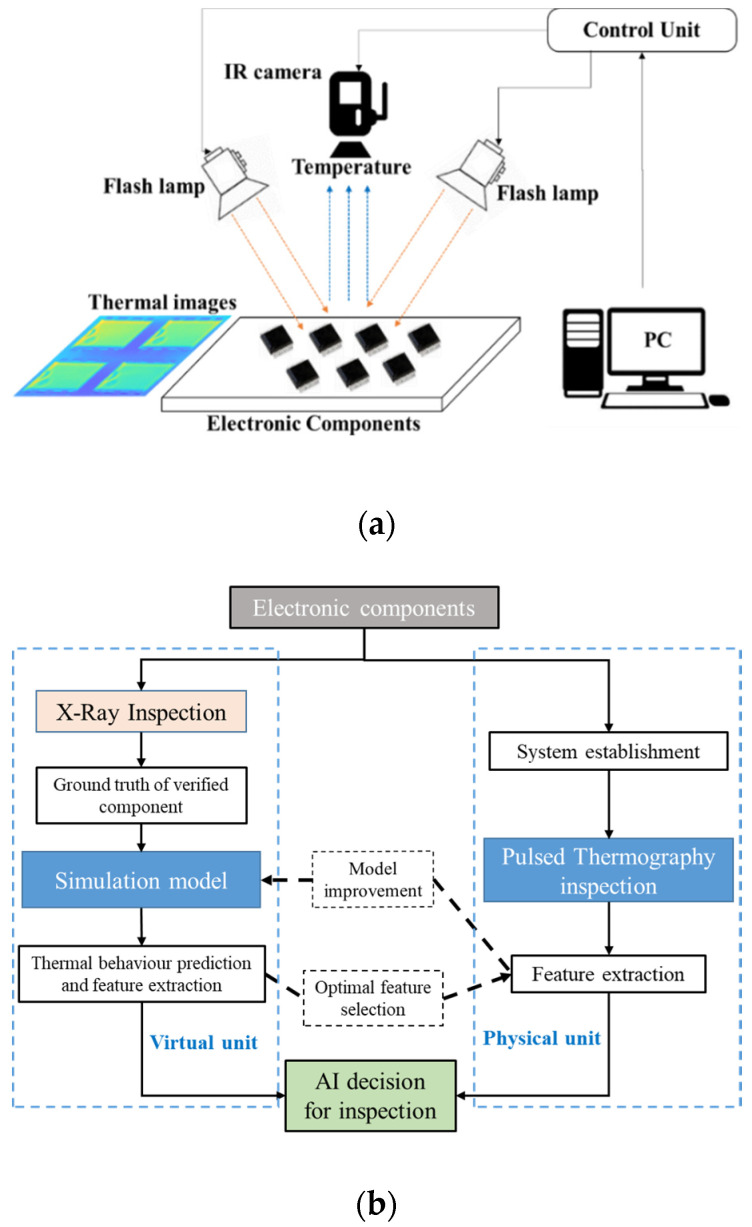
The methodology framework. (**a**) The principal of pulsed thermography; (**b**) The proposed dual-path strategy for electronic components inspection.

**Figure 4 sensors-20-05013-f004:**
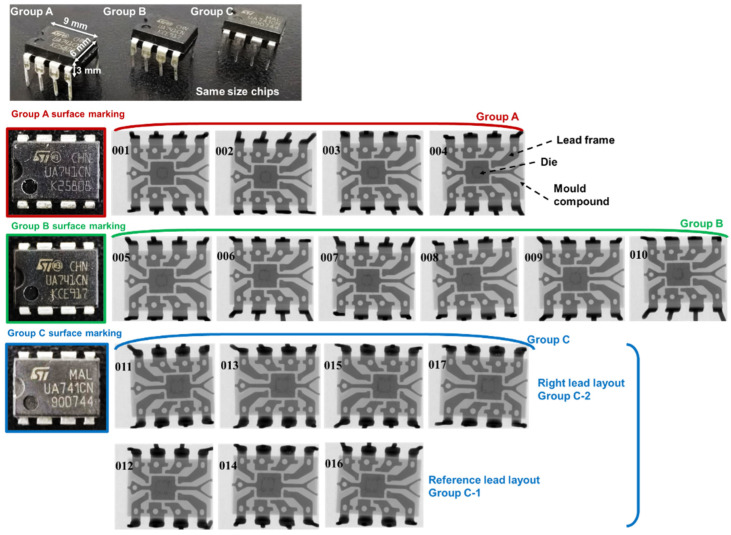
The dimensions, digital images and X-ray images of three groups of chips, where the differences in surface and inner structure can be clearly observed.

**Figure 5 sensors-20-05013-f005:**
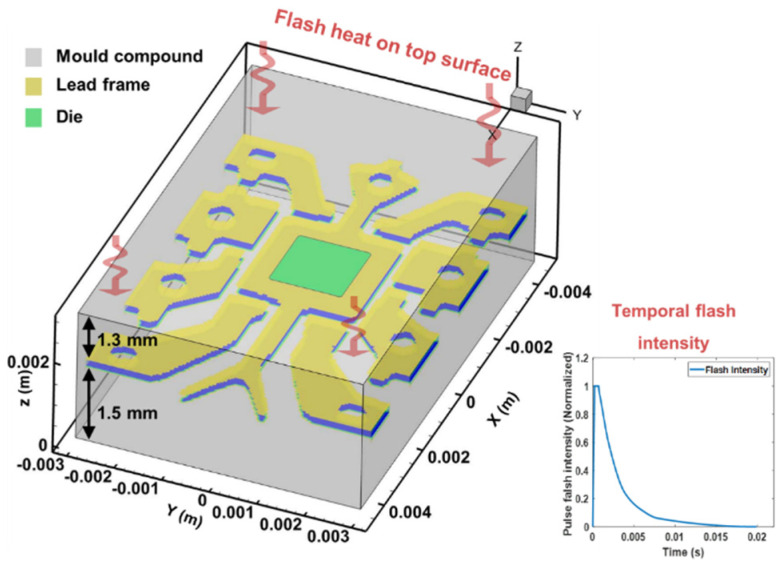
FEM modelling based on the X-Ray image.

**Figure 6 sensors-20-05013-f006:**
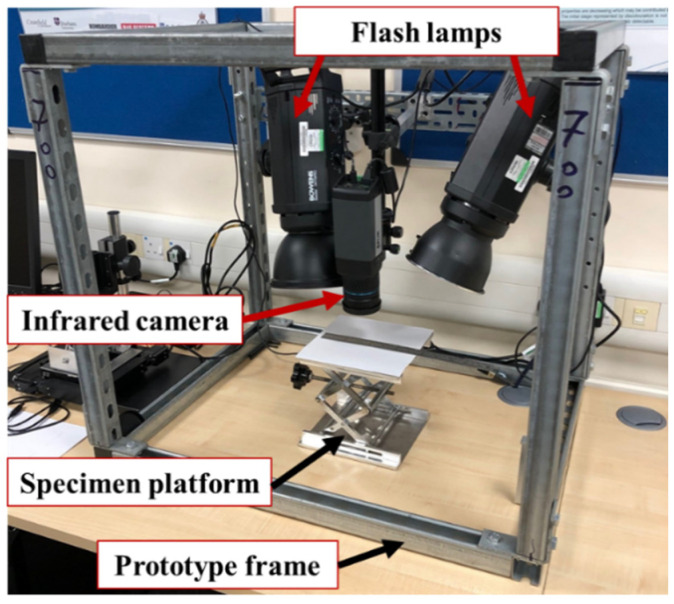
Experimental testing unit.

**Figure 7 sensors-20-05013-f007:**
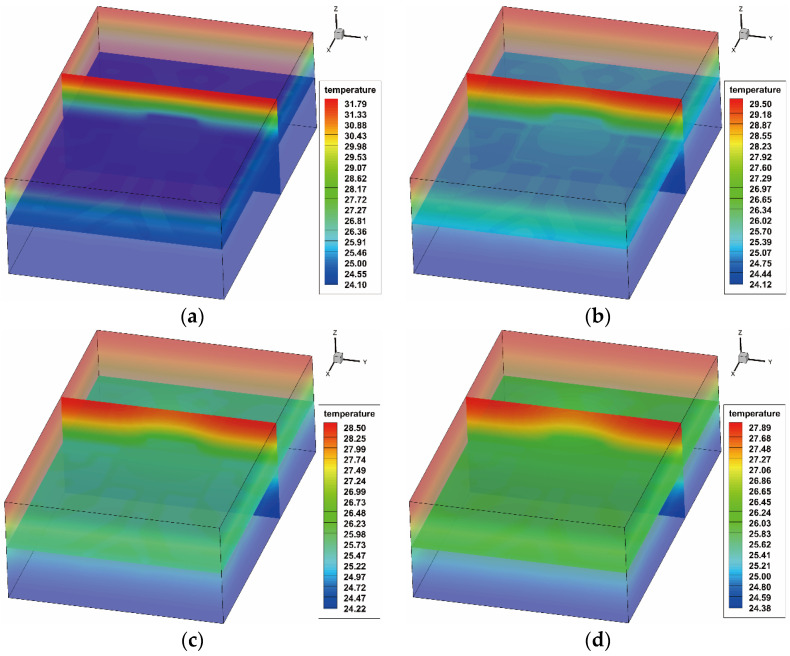
Temperature distribution after the flash; (**a**) 0.04s after the flash; (**b**) 0.08s after the flash; (**c**) 0.12s after the flash; (**d**) 0.16s after the flash.

**Figure 8 sensors-20-05013-f008:**
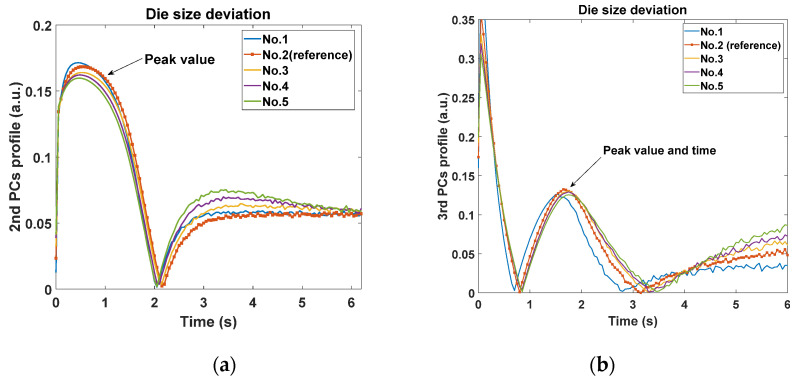
Comparison of features with different die sizes; (**a**) the 2nd principal component profiles; (**b**) the 3rd principal component profiles.

**Figure 9 sensors-20-05013-f009:**
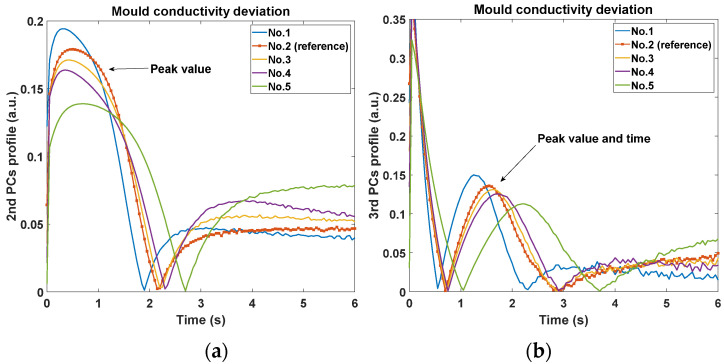
Comparison of features with different values of mould conductivity; (**a**) the 2nd principal component profiles; (**b**) the 3rd principal component profiles.

**Figure 10 sensors-20-05013-f010:**
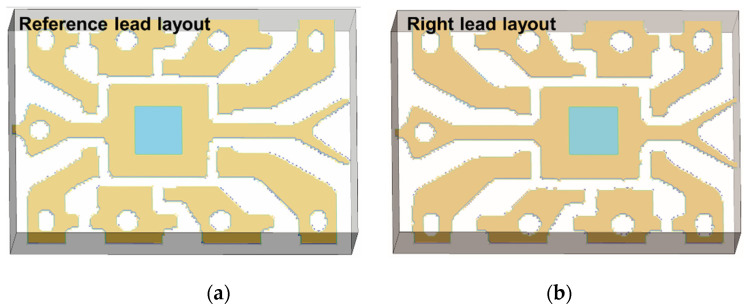
Comparison of features with different lead frame layouts; (**a**) model of the reference lead layout; (**b**) model of the right lead layout; (**c**) the 2nd principal component profiles; (**d**) the 3rd principal component profiles.

**Figure 11 sensors-20-05013-f011:**
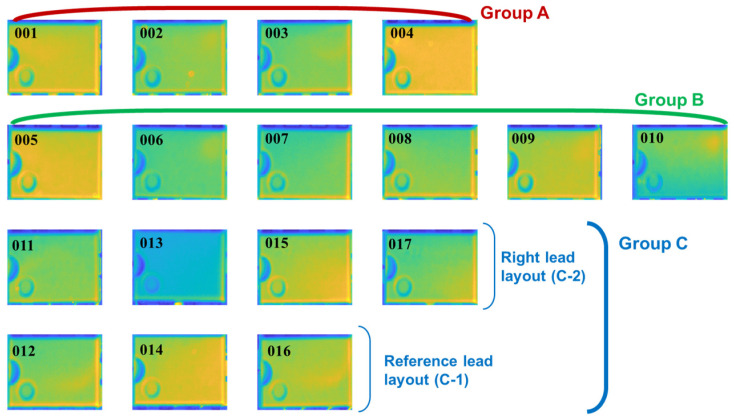
The first temperature frame (0.04 s) after the flash.

**Figure 12 sensors-20-05013-f012:**
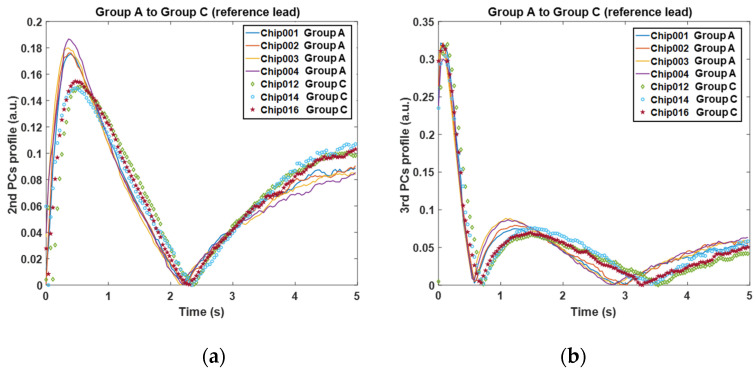
Feature results of the 1st verification comparison; (**a**) the 2nd principal component profiles; (**b**) the 3rd principal component profiles.

**Figure 13 sensors-20-05013-f013:**
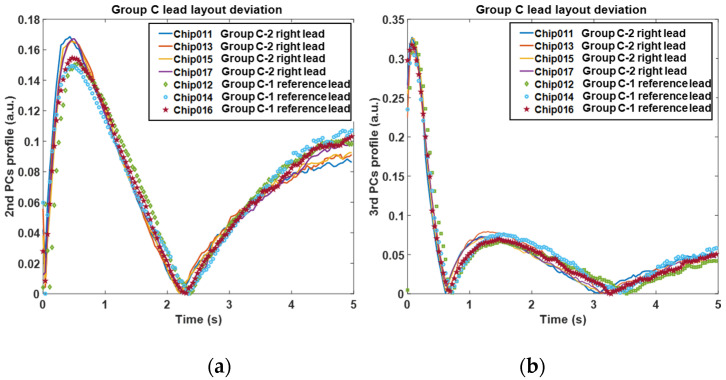
Feature results of the 2nd verification comparison; (**a**) the 2nd principal component profiles; (**b**) the 3rd principal component profiles.

**Figure 14 sensors-20-05013-f014:**
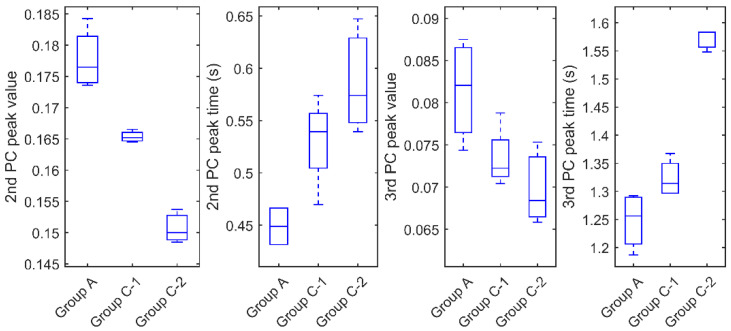
Quantitative comparisons of peak features among three groups in the experiments (Group C-1: Chips with Reference lead layout in Group C; C-2: Chips with Right lead layout in Group C).

**Figure 15 sensors-20-05013-f015:**
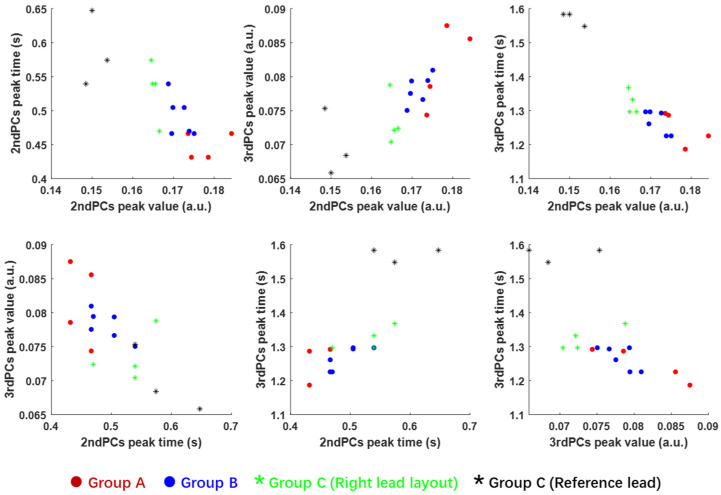
Scatter plots of different feature combination for Group A, B and C.

**Figure 16 sensors-20-05013-f016:**
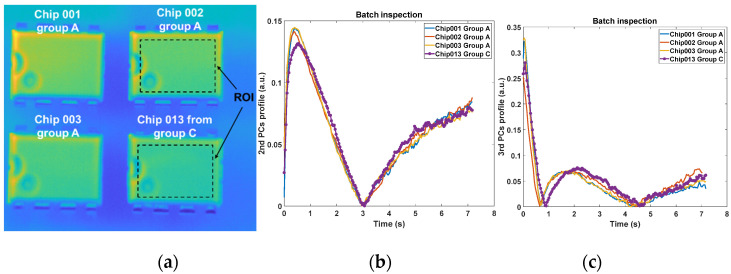
Results of the batch-chip inspection where 3 chips from Group A and 1 chip from Group C are inspected at the same time. (**a**) the first temperature frame (0.04 s) after the flash; (**b**) the 2nd principal component profiles; (**c**) the 3rd principal component profiles.

**Table 1 sensors-20-05013-t001:** Material thermal properties applied in the simulation.

Component (Material Type)	Conductivity	Volumetric Heat Capacity
Mould compound (Resin)	5.40 W/(m⋅K)	1.33 × 10^6^ J/K/m^3^
Lead frame and pins (Copper)	3.85 × 10^2^ W/(m⋅K)	3.45 × 10^6^ J/K/m^3^
Die (Silicon)	1.30 × 10^2^ W/(m⋅K)	1.65 × 10^6^ J/K/m^3^

**Table 2 sensors-20-05013-t002:** Model attribute deviations.

Deviation type: Only Die Size Deviation (mm^2^)					
No. 1: 0.49 (30%)	No. 2: 1.51 (reference)	No. 3: 2.25 (150%)		No. 4: 2.90 (200%)	No. 5: 3.77 (250%)
**Deviation Type: Only Mould Material Conductivity Deviation (W/(m⋅K))**					
No. 1: 5.94 (110%)	No. 2: 5.40 (reference)	No. 3: 4.86 (90%)		No. 4: 4.32 (80%)	No. 5: 3.78 (70%)
**Deviation Type: Only Lead Frame Layout Deviation**					
Reference lead (Group A)			Right lead layout (‘011’ in Group C)		

**Table 3 sensors-20-05013-t003:** Classification accuracy using the 2nd PC peak value and 3rd PC peak time.

	Two Classes		Three Classes	Four Classes
	Group A vs. Group B	Group A vs. Group C	Group A vs. Group B vs. Group C	Group A vs. Group B vs. Group C-1 vs. Group C-2
KNN	70%	100%	88.2%	88.2%
LD	70%	100%	82.4%	82.4%
SVM	70%	100%	82.4%	76.5%

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
