# Peer review of "A Novel Inspection Technique for Electronic Components Using Thermography (NITECT)"

_sensors, 2020, doi:10.3390/s20175013_

Round 1
Reviewer 1 Report
This paper introduced a screening tool for electronic chips based on the thermography in a non-destructive, time effective and automatic manner to complement the existing inspection capabilities. The paper is interesting, informative and well written. Suggest acceptance with minor revision.
Comments:
1) More details about the FEM modeling is necessary as it was supposed as a "digital twin" of the chips.
2) More reference papers are necessary about the thermograph and related NDT techniques. In addition, papers under review are not appropriate to be included in the reference.
Author Response
This paper introduced a screening tool for electronic chips based on the thermography in a non-destructive, time effective and automatic manner to complement the existing inspection capabilities. The paper is interesting, informative and well written. Suggest acceptance with minor revision.
Comments 1.1: More details about the FEM modeling is necessary as it was supposed as a "digital twin" of the chips.
Reply: More details about the FEM model parameters have been added in Section 2.2.
Comments 1.2: More reference papers are necessary about the thermograph and related NDT techniques. In addition, papers under review are not appropriate to be included in the reference.
Reply: Two references [32], [33] have been supplemented in Section 2.2. The under-review reference has also been removed.
Reviewer 2 Report
A Novel Inspection Technique for Electronic Components using Thermography (NITECT). This is a good and interesting paper for thermography NDT. Just some minor comments
- Introduction, the review paper about thermography NDT could be added: Optically and Non-optically Excited Thermography for Composites: A review, Infrared Physics & Technology, 2016, 75:26-50; Recent Advances in Active Infrared Thermography for Non-Destructive Testing of Aerospace Components
- Some symbols in text should be italic;
thanks
Author Response
A Novel Inspection Technique for Electronic Components using Thermography (NITECT). This is a good and interesting paper for thermography NDT. Just some minor comments
Comments 2.1: Introduction, the review paper about thermography NDT could be added: Optically and Non-optically Excited Thermography for Composites: A review, Infrared Physics & Technology, 2016, 75:26-50; Recent Advances in Active Infrared Thermography for Non-Destructive Testing of Aerospace Components;
Reply: The suggested references have been reviewed and added as [20] and [21].
Comments 2.2: Some symbols in text should be italic;
Reply: The appropriate font in typo mistakes have been checked and modified.
Reviewer 3 Report
- The title needs to be modified with a title that can infer the content of the study.
- The abstract needs to be changed to a condensed expression that includes the summary of the study, the experimental procedure, and the results, and also needs to be reorganized concisely.
- In Figure 2, it is necessary to explain in the order of (a), (b) or change the order of (a) and (b).
- For the first terms such as PT, ECs, etc. in Figure 2, it is necessary to indicate the full name.
- In the case of an introduction, it is necessary to rewrite with a concise description of the content including the research background and necessity.
- Introduction and Methodology sections are written interchangeably. It is necessary to organize this, and rewrite it in a concise and compact sentence.
- Figure 4 needs to be rearranged so that groups A, B, and C can be expressed consistently to increase readability.
- Figure 4 requires an additional explanation of the differences in each result.
- In Figure 5, it is necessary to add a detailed explanation for the method of FEM modeling.
- No logical explanation for the result obtained in Figure 7 was found
- The background of the sudden appearance of Table 2 and no explanation of it was found.
- It is necessary to provide the explanation on page 11 logically.
- In Figures 8, 9, and 10, only the presentation of the results, but no consideration of the results was found.
- No explanation of the physical meaning of the results in Figure 11 was found
- The results of Figures 12, 13 and 14 are not known by which consistency are described.
- What does the result of Figure 16 mean? This part should be provided in the 3. Results session.
- The conclusion does not fully explain the research results.
- Overall, the paper is edited in a too lengthy composition, so it needs to be corrected.
- Terms such as Machine Learning, AI, and Digital Twin are used in the text, but lack a specific explanation, and lack of sufficient explanation of the linkages, reasons for introduction, and appropriate results among these various technologies, which are causing a lack of understanding among readers.
Author Response
Comments 3.1: The title needs to be modified with a title that can infer the content of the study.
Reply: The title has been changed to “Anomaly inspection of electronic components using pulsed thermography” to better represent the context of this study.
Comments 3.2: The abstract needs to be changed to a condensed expression that includes the summary of the study, the experimental procedure, and the results, and also needs to be reorganized concisely.
Reply: As suggested, the abstract has been rewritten.
Comments 3.3: In Figure 2, it is necessary to explain in the order of (a), (b) or change the order of (a) and (b).
Reply: Figure 2 has been reorganized with the associated caption.
Comments 3.4: For the first terms such as PT, ECs, etc. in Figure 2, it is necessary to indicate the full name.
Reply: All acronyms have been checked and revised to ensure they have the full name at the first-time appearance.
Comments 3.5: In the case of an introduction, it is necessary to rewrite with a concise description of the content including the research background and necessity.
Reply: The introduction has been carefully modified. The necessity, advantages of PT inspection for EC, and the research reviews have been refined and marked in the Introduction.
Comments 3.6: Introduction and Methodology sections are written interchangeably. It is necessary to organize this, and rewrite it in a concise and compact sentence.
Reply: As suggested, the description of the principle of PT for EC inspection (as shown below) associated with Figure 3 has been rewritten and moved into the first paragraph of the Methodology.
Comments 3.7: Figure 4 needs to be rearranged so that groups A, B, and C can be expressed consistently to increase readability.
Reply: Figure 4 has been modified and rearranged for clearer readability.
Comments 3.8: Figure 4 requires an additional explanation of the differences in each result.
Reply: The caption of Figure 4 has been rewritten to explain the difference. A more detailed explanation can be found in the second paragraph of Section 2.1.
Comments 3.9: In Figure 5, it is necessary to add a detailed explanation for the method of FEM modeling.
Reply: More details of FEM modelling have been added in the first and the last paragraphs of Section 2.2.
Comments 3.10: No logical explanation for the result obtained in Figure 7 was found.
Reply: Figure 7 presents the temperature distribution at a series of time stamp after the flash to demonstrate the feasibility of simulation. The visualization presents the quality of response calculation and fineness of the discritized model, which is a common presentation in thermal simulation studies [1, 2].
[1] Lopez, F., de Paulo Nicolau, V., Ibarra-Castanedo, C. and Maldague, X., 2014. Thermal–numerical model and computational simulation of pulsed thermography inspection of carbon fiber-reinforced composites. International Journal of Thermal Sciences, 86, pp.325-340.
[2] Li, T., Almond, D. and Rees, D., 2011. Crack imaging by scanning laser-line thermography and laser-spot thermography. Measurement Science and Technology, 22(3), p.035701.
Comments 3.11: The background of the sudden appearance of Table 2 and no explanation of it was found. It is necessary to provide the explanation on page 11 logically.
Reply: Some explanations been added to the second paragraph in Section 3.1.
Comments 3.12: In Figures 8, 9, and 10, only the presentation of the results, but no consideration of the results was found.
Reply: The discussion of Figure 8-10 is provided in the third paragraph of Section 3.1. To better clarify the results, the following explanation has been added to this paragraph:
“Viewing the feature trends of die size, it represents that the bigger die has the smoothing effect of the 2nd PC peak and delay the 3rd PC peak. For the mould material deviation, the feature trend indicates the lower conductivity mould will maintain the heat flow longer resulting in similar smoothing and delay effect.”
Comments 3.13: No explanation of the physical meaning of the results in Figure 11 was found.
Reply: The physical reason has been added in the first paragraph of Section 3.1 as following. “The reason is mainly caused by the uncertainty of the surface emissivity of samples which results in different temperature rise after the pulse. Besides, the uncertainty of surface homogeneity led to diversity of thermal spatial distribution in the first frame.”
Comments 3.14: The results of Figures 12, 13 and 14 are not known by which consistency are described.
Reply: The legends of Figure 12, 13 and 14 have been modified by adding the group number for clearer readability. The feature trend of three types of deviation in these experimental results is consistent with the simulation results and in return proves the effectiveness of the simulation unit.
Comments 3.15: What does the result of Figure 16 mean? This part should be provided in the 3. Results session.
Reply: All results shown in Section 3 are based on the inspection of single chip. The result of Figure 16 is a feasibility validation for inspection of a batch of ECs, which will improve the inspection speed. It demonstrates the proposed feature is capable to screen the different groups at the same time. However, it should be noted for the current prototype, the camera view sight cannot contain too many chips (maximum 4). Therefore, we distribute it as one part of future extendibility discussion.
Comments 3.16: The conclusion does not fully explain the research results.
Reply: The conclusion has been rewritten in the revised version.
Comments 3.17: Overall, the paper is edited in a too lengthy composition, so it needs to be corrected.
Reply: According to the above comments suggested by the reviewer, the manuscript is more concise.
Comments 3.18: Terms such as Machine Learning, AI, and Digital Twin are used in the text, but lack a specific explanation, and lack of sufficient explanation of the linkages, reasons for introduction, and appropriate results among these various technologies, which are causing a lack of understanding among readers.
Reply: These terms have been explained and linked in the revised version.
Reviewer 4 Report
The authors presented a method to detect Unverified or counterfeited electronic components based on pulsed thermography combined to FEM and X-ray inspection as ground truth. Thermographic data is processed by PCT and the 2nd and 3rd PCs are used to classify samples into one of two groups (sound or damaged (either malfunctioning/broken, or geometrical deviations).
FEM results were in agreement with experimental results, and proved that the combination of particular features (2nd and 3rd PCs peak values, peak times) could be used to classify components.
As a suggestion, authors may consider to also exploit in the future the spatial information from the EOFs, for instance, for the determination of the lead frame layouts.
The manuscript is interesting and well written, consider the following two minor suggestions :
Abstract, please write: "...caused by equipment and samples..."... instead of "caused equipment and samples.."
Page 4, last two lines: Please write, "The inspection results can validate not only the performance of the simulation,...." instead of "The inspection results can not only validate the performance of the simulation,..."
Author Response
The authors presented a method to detect Unverified or counterfeited electronic components based on pulsed thermography combined to FEM and X-ray inspection as ground truth. Thermographic data is processed by PCT and the 2nd and 3rd PCs are used to classify samples into one of two groups (sound or damaged (either malfunctioning/broken, or geometrical deviations). FEM results were in agreement with experimental results, and proved that the combination of particular features (2nd and 3rd PCs peak values, peak times) could be used to classify components.
Comments 4.1: As a suggestion, authors may consider to also exploit in the future the spatial information from the EOFs, for instance, for the determination of the lead frame layouts.
Reply: This constructive comment is much appreciated. The capability of EOFs for spatial information of lead frame visualisation will be carefully investigated. It has been added in the future work in the conclusions, as following: “In feature selection approach, the EOFs in PCA could contain more spatial information of the ECs internal structure like lead and die layout which could enhance the inspection visualisation. It will be investigated carefully in future.”
Comments 4.2: The manuscript is interesting and well written, consider the following two minor suggestions :1) Abstract, please write: "...caused by equipment and samples..."... instead of "caused equipment and samples..". 2) Page 4, last two lines: Please write, "The inspection results can validate not only the performance of the simulation,...." instead of "The inspection results can not only validate the performance of the simulation,..."
Reply: Thank you for your comments about the description. Both suggested changes have been adopted.
Round 2
Reviewer 3 Report
Appropriate actions have been taken against the previous comments.